# Utility of iPSC-Derived Cells for Disease Modeling, Drug Development, and Cell Therapy

**DOI:** 10.3390/cells11111853

**Published:** 2022-06-06

**Authors:** Martin W. Nicholson, Chien-Yu Ting, Darien Z. H. Chan, Yu-Che Cheng, Yi-Chan Lee, Ching-Chuan Hsu, Ching-Ying Huang, Patrick C. H. Hsieh

**Affiliations:** Institute of Biomedical Sciences, Academia Sinica, Taipei 115, Taiwan; mawnicho@ibms.sinica.edu.tw (M.W.N.); chienyu@ibms.sinica.edu.tw (C.-Y.T.); darien.czh@ibms.sinica.edu.tw (D.Z.H.C.); criss1125@ibms.sinica.edu.tw (Y.-C.C.); paul84114@gmail.com (Y.-C.L.); gingerhsu0904@gmail.com (C.-C.H.); jenniferhuang0820@gmail.com (C.-Y.H.)

**Keywords:** induced pluripotent stem cells, cell therapy, cardiomyocytes, neurons

## Abstract

The advent of induced pluripotent stem cells (iPSCs) has advanced our understanding of the molecular mechanisms of human disease, drug discovery, and regenerative medicine. As such, the use of iPSCs in drug development and validation has shown a sharp increase in the past 15 years. Furthermore, many labs have been successful in reproducing many disease phenotypes, often difficult or impossible to capture, in commonly used cell lines or animal models. However, there still remain limitations such as the variability between iPSC lines as well as their maturity. Here, we aim to discuss the strategies in generating iPSC-derived cardiomyocytes and neurons for use in disease modeling, drug development and their use in cell therapy.

## 1. Introduction

Between 2009 and 2018, the cost of bringing a new drug from conception to the market was approximately USD 985 million [1]. Even though rigorous testing is undertaken during the development stages, 90% of drugs fail clinical trials [2]. This is largely due to the use of irrelevant cell culture systems or animal models that do not accurately model the human system. Only during clinical trials is the human context considered, when efficacy issues and adverse drug reaction are observed.

Human induced pluripotent stem cells (iPSCs) and their derivatives provide a human-relevant cell source and thus have been employed for their use in a variety of scientific fields including drug discovery, toxicity studies, and disease modeling. The capacity of iPSCs for the scalable production of diverse cell types benefits researchers by reducing obstacles with procedures such as apheresis, which involves separating specific components from the blood and retuning the remaining components to the donor’s circulation, whilst obtaining human cells [3]. Various cell models built with iPSC derivatives have been shown to resemble in vivo or primary cell counterparts at transcriptional, cellular, and functional levels whilst leaving little interference to the genetic content [4,5,6,7]. Cardiovascular and neurological diseases are among the top leading causes of death worldwide and are expected to increase with the ageing population. Jointly, iPSCs act as an ideal cell source for disease, drug, and cell therapy studies. In this review we aim to discuss the recent advances in iPSC-derived cardiovascular and neurological disease models used in drug discovery and their potential use in cell therapy.

## 2. Drug Discovery Using iPSC-Derived Disease Models

The main aims of generating a disease model are to elucidate the function and molecular mechanisms of the disease, discover a new drug target, and/or develop a new therapy (Figure 1). Traditionally, animal models, primary cells, embryonic stem cells (ESC), and immortalized cell lines derived from tumor cells are used to model a variety of human diseases [8,9,10,11]. Although many of these models offer specific advantages, the underlying issue is that they often do not accurately model human diseases. Furthermore, ESCs raise an inextricable ethical issue, which largely hinders their utility [12,13]. The physiological and genetic differences between human and animals leads to potential differences in terms of function, mechanisms, and signaling pathways [14,15]. Tumorous cell lines, although able to proliferate, are often incorporated with unwanted genetic or chromosomal abnormalities, which might induce interference to the resulting outcomes [4,5]. Primary cells serve as a relevant source for patient-specific modeling and drug testing, however, the difficulty involved in their harvesting and culturing means they have limited life span and often a low yield. With reproducible reprogramming and differentiation protocols, human iPSCs stand out as a potent and relevant cell source for modelling human diseases. iPSCs and their derivatives provide models that are self-renewing, genetically preserved and interference-free. Models built with iPSC derivatives also demonstrate functional and phenotypic traits comparable to their human in vivo parallel [4,5,6]. As well as modelling diseases, iPSC-derivatives provide a relevant platform for drug discovery and toxicity screening. They are seen in a variety of studies generating screening platforms of different cell types and for diverse purposes.

### 2.1. Cardiomyocytes

To date, there has been an extensive list of cardiovascular models generated from patient-derived iPSCs such as long QT syndrome (LQTS), hypertrophic cardiomyopathy, Leopard syndrome, and arrhythmogenic right ventricular cardiomyopathy/dysplasia [1,2,3,4]. LQTS is an inherited cardiac disease with a prolongation of the QT interval which can cause arrhythmia. Previous studies using iPSC-derived cardiomyocytes from type-1 LQTS patients revealed that the mutation in *KCNQ1* caused potassium ion channel dysfunction resulting in sarcolemmal deficiency [5,6]. Type-2 LQTS patients’ iPSC-derived cardiomyocytes showed that a mutation in *KCNH2* causes action-potential-duration prolongation by reducing the cardiac potassium current IKr and dysfunction of the sodium ion channel [7,8,9]. Since patients’ iPSC-derived cardiomyocytes can mimic cardiac ion channel diseases, Matsa et al. used type-2 LQTS patients iPSC-derived cardiomyocytes to show that treatment with experimental potassium channel enhancers, nicorandil and PD118057, can cause action potential shortening [10]. Using iPSC-derived cardiomyocytes from type-3 LQTS patients with *SCN5A* mutation revealed abnormal calcium transients with diminished INa current density [11]. They also showed that mexiletine can inhibit INa and shorten the QT interval pathological effect of the mutated sodium channel [12]. Notably, by employing iPSC-derived cardiomyocytes, they found that new mexiletine analogues can enhance potency and selectivity for INa [13], discovering a potential new therapy for type-3 LQTS.

In addition, cardiomyopathy is a severe disease of the heart muscle with structural and functional abnormalities, which can lead to heart failure. The heart causing diastolic dysfunction, and ventricular arrhythmogenesis are the main pathological features of hypertrophic cardiomyopathy (HCM). Research using iPSC-derived cardiomyocytes from a patient with a mutation in *MYH7* recapitulated HCM phenotypes including enlarged cellular size, disrupted sarcomere structures, contractile arrhythmia, and dysregulation of Ca^2+^ cycling. This iPSC-derived model played an important role in understanding the effect of MYH7 mutation [14,15]. Drug screening was also performed to identify a new therapy for HCM using four different calcium channel blockers. Results showed that verapamil has the strongest inhibition on iPSC-derived cardiomyocyte contractile function and decreased expression of contraction-related genes, such as *MYH* and *troponin I* [16]. Dilated cardiomyopathy (DCM) is another type of cardiomyopathy that displays an enlarged cavity of the heart, wall thinning, and impaired systolic function. iPSC-derived cardiomyocytes from DCM patients revealed that several different mutations including *titin* [17], *troponin T* [18], *RBM20* [19], and *lamin* [20,21,22] are related to DCM through different molecular mechanisms. For DCM therapy, all-trans retinoic acid was found to rescue DCM iPSC-CMs with *RBM20* mutations [19]. Siu et al. found MEK1/2 inhibitors, U0126 and selumetinib (AZD6244) that attenuate proapoptotic phenotype of DCM iPSC-CMs with *lamin* mutation [20]. Healthy iPSC-derived cardiomyocytes also play an important role in toxicity assessment. Sharma et al., 2017 performed a high-throughput toxicity screen assessing the cardiotoxicity of small molecule tyrosine kinase inhibitors. As these small molecules are beneficial in treating cancer, they generated a “cardiac safety index” to reflect the safety of existing tyrosine kinase inhibitors [23].

One main issue related to iPSC-derivatives is their questionable maturity. Miki et al., 2021 identified two compounds, ERRγ agonist and SKP2 inhibitor, that play a role in enhancing CM maturation [24]. Manifestation of the TNNI1 to TNNI3 transition, indicating maturation of cardiomyocytes, was confirmed following the treatment with the two compounds. This platform was essential in identifying potential targets to generate mature hiPSC-CMs in order to compensate for the lack of maturity commonly seen in many current hiPSC-CM models [24]. The co-culture models of cardiomyocytes and mesenchymal stem cells have also been shown to improve cardiomyocyte functionality as mesenchymal stem cells secrete soluble growth factors, enhancing cardiomyocyte maturity [25]. Similarly, co-culture with cardiac fibroblasts or endothelial cells has also been shown to influence cardiomyocyte maturation, hypertrophy, function and gene expression [26,27,28]. Therefore, co-culture models may represent a more relevant model system for the study of cardiovascular disease as well as drug screening. However, as multiple cell types are present, data analysis becomes more complicated, therefore, more stringent and throughout analysis would need to be performed. Nevertheless, the data obtained may be more relevant than the use of a homogenous cell culture.

### 2.2. Neurons

#### 2.2.1. Neurodegenerative Diseases

A growing health concern is diseases related to neurodegeneration, which is characterized by a progressive loss of neurons. This loss of neurons influences movement, speech, memory, and cognitive ability. Alzheimer’s disease (AD) is the most common type of neurodegenerative disease that affects memory and behavior. IPSC-derived neurons display many of the cellular phenotypes found in AD patients which are associated with amyloid precursor protein (APP) [29], presenilin [30], or SORL1 mutation [31], which are involved in amyloid precursor protein processing; dysfunction of these proteins could lead to impaired γ-secretase activity, endoplasmic reticulum and oxidative stress with tau protein hyperphosphorylation, and amyloid β peptide (Aβ) accumulation. One study employing AD patient iPSC-derived neurons showed that treatment with docosahexaenoic acid could alleviate the associated stress responses [32]. Another study showed specific anti-Aβ compounds could reduce Aβ plaque deposition in patient iPSC-derived cortical neurons using a drug screening platform [33]. Bassil et al., 2021, generated a model of AD which consisted of neurons, astrocytes, and microglia. This model presents AD-like hallmarks such as neuronal loss induced by p-Tau. They then employed this model to screen for potential therapeutics for the treatment of AD. The screening capability was examined by validating nine compounds, and the authors have successfully demonstrated a protecting effect on neurons with compounds known to inhibit Tau phosphorylation [34].

Parkinson’s disease (PD) is the second most prevalent neurodegenerative disease after AD with a progressive loss of ventral midbrain dopaminergic neurons (vmDAns). PD symptoms include bradykinesia, impaired balance, and rigid muscle. SNCA triplication patient iPSC-derived dopaminergic neurons showed α-synuclein protein over accumulation which was consistent with pathological phenotype in PD patient [35]. PD patient iPSC-derived dopaminergic neurons harboring *Parkin* mutation displayed decreased microtubule stability, increased oxidative stress, reduced dopamine uptake, and increased spontaneous dopamine release [36,37]. Moreover, PD patient iPSC-derived dopaminergic neurons harboring *LRRK2* and *PINK1* mutations showed impaired axonal outgrowth and deficient autophagic vacuole clearance which cause abnormal α-syn accumulation [38]. Several researchers tried to figure out potential therapeutic candidates for PD. The pharmacological capability of Coenzyme Q10, rapamycin, and LRRK2 kinase inhibitor GW5074 were found in iPSC-derived neural cells from PD patients with *LRRK2* and *PINK1* mutations [38].

Amyotrophic lateral sclerosis (ALS) is a kind of neurodegenerative disease that affects motor neurons. Familial ALS patient iPSC-derived motor neurons with TDP-43 mutation showed decreased voltage-activated Na+ and K+ currents and increased vulnerability [39]. Fused in sarcoma (FUS) [40,41] and superoxide dismutase 1 (SOD-1) [42,43] are also linked to ALS. ALS patient iPSC-derived motor neurons with these mutants revealed increased proapoptotic factors, increased oxidative stress, reduced mitochondrial function, and inhibited oxidative damage repair. Ropinirole was identified as a potential therapeutic candidate in ALS patient iPSC-derived motor neurons with TDP-43 and FUS [44]. However, the majority of ALS is sporadic without a known genetic link. Recently, a comparison of sporadic and familial ALS iPSC-derived motor neurons revealed that ELAVL3 misexpression in motor neurons becomes a new ALS hallmark in ALS therapeutic drug discovery [45].

#### 2.2.2. Neurodevelopmental Diseases

Neurodevelopmental disorders are complex and diverse diseases and the molecular progression is difficult to track due the lack of an in vitro model. The use of iPSC-derived neurons has also greatly enhanced our understanding of psychiatric disorders such as autism spectrum disorder (ASD). ASD is a complex development disease in the brain with complex etiology or genetic pathology, characterized by restricted/repetitive behaviors and social communication disturbance. Transcriptome and gene network analysis revealed that ASD organoids display a decreased cell cycle, synaptic overgrowth, and overproduction of GABAergic neurons [46]. In iPSC-derived neurons from an ASD patient carrying TRPC6 mutation, neurons showed abnormal morphology, such as reduced dendritic arborization, fewer dendritic spines, and synapses [47]. Interestingly, treatment with insulin-like growth factor 1 (IGF-1) or hyperforin, a TRPC6-specific agonist, is able to rescue the neuronal abnormalities [47]. However, increased dendrite length, dendrite complexity, synapse number, and frequency of spontaneous excitatory postsynaptic currents were observed in iPSC-derived neurons from an ASD patient carrying SHANK2 mutation. The dendrite length increase was exacerbated by IGF1 and suppressed by DHPG treatment [48]. These results suggest a bidirectional capability of IGF-1 in ASD therapeutic drug discovery.

Patient iPSC-derived neurons provide a unique opportunity to uncover the molecular mechanisms involved in many neurodevelopmental disorders. Spinal muscular atrophy (SMA) is a disease generated by several different hereditary neurodevelopmental mutations with a progressive loss of motor neurons. The most common form of SMA is caused by defects in the survival motor neuron (SMN1) expression. SMA iPSC-derived motor neurons showed a higher vulnerability with reduced dendritic and axonal length. RNA analysis from SMA iPSC-derived motor neuron demonstrates hyperactivation of the endoplasmic reticulum (ER) stress pathway due to activated unfolded SMN [49]. SMA iPSC-derived neurons are not only used in the pathogenic investigation but also in therapeutic discovery. Treatment with valproic acid, cyclic tetrapeptide histone deacetylase inhibitors, and thyrotropin-releasing hormone (TRH) analog could enhance SMN protein expression to improve motor neuron defects [50]. Bringing drugs from the lab to clinical trials is the end goal of many drug screening studies. Ohuchi et al., 2006, first used SMA patient iPSC-derived motor neurons and astrocytes to show the role of the thyrotropin-releasing hormone analog for the treatment of SMA, which has been taken to clinical trials and showed the potential efficacy of this drug for the treatment of SMA [51].

Kaufmann et al., 2015, set out to find a therapeutic option for the treatment of fragile X syndrome. They first generated a reporter cell line for Fmr1 expression. They performed a high-content drug screen of 50,000 compounds using neuronal progenitor cells and identified several compounds that induced expression of the fragile X mental retardation protein [52].

### 2.3. Organoids

Initially, most differentiation protocols were developed using traditional 2D culture conditions. Although these cultures are easier to maintain and easier to assay, they lack complex 3D structures. This hampers their ability to accurately model human biology and pathophysiology. Therefore, it has led researchers to develop differentiation protocols that allow for 3D culture generation which better recapitulates the environment found at the human tissue and organ-levels. These 3D “organoids” are often found to have instructive morphogenetic cues resulting in self organization similarly found in vivo [53,54]. These models have been particularly important in the search for anticancer compounds as they can predict patient outcomes in phase 1/2 clinical trials for gastrointestinal cancer [55].

#### 2.3.1. Cardiac Organoids

Cardiac organoids have played an important role in understanding cardiac diseases such as arrhythmia and cardiac repair after injury [56,57]. One group has generated a highly complex cardiac organoid that includes a central, void chamber [58]. Another group, which previously developed a high-throughput bioengineered human cardiac organoid platform [59], has generated organoids for the purpose of identifying pro-proliferative compounds, which was followed by validation and functional screening to identify hit compounds with side effects on contractility [56]. They used these organoids to further validate the functional pathways involved in the proliferative effects. Self-assembling human cardiac organoids have also been used to show the role of BMP4 and Activin A to improve heart organoid chamber formation and vascularization [60]. Another group has developed a 96-well format for functional screening of cardiac organoid viability, function, and maturation. They screened over >10,000 organoids to identify optimal maturation conditions with the aim for them to be used in future high-throughput drug screening studies [61]. Cardiac organoids have also been used to screen a panel of environmental toxins, showing the utility of these models for the use of toxicity screening [62].

#### 2.3.2. Neuronal Organoids

Neuronal organoids have provided a great understanding to many neurological and developmental diseases. They have also contributed greatly to the understanding of many psychiatric disorders such schizophrenia and autism spectrum disorder [63,64]. Using cerebral organoids, Stachowiak et al., 2017, showed that schizophrenia organoids displayed an abnormal pattern of Ki67+ neural progenitor cells from the ventricular, intermediate, and cortical zones not observed in 2D cultures [64]. Another study using 100 day cerebral cortical organoids showed that organoids derived from patients with 22q11.2 deletion syndrome had deficits in spontaneous neuronal activity and calcium signaling. The authors showed that antipsychotics could restore the functional defects found in 22q11.2 deletion syndrome providing further justification for the use of iPSC-derived organoids as a relevant model for disease modeling [65]. In addition to acting as a model in order to further our understanding of various neurological diseases, organoids have begun to play an important role in drug screening. The 2015/16 Zika virus outbreak sparked the search for anti-ZIKV compounds. Zhou et al., 2017 employed high-content screening of hPSC-forebrain organoids to screen >1000 FDA approved drug candidates. They were able to identify hippeastrine hydrobromide as a compound that not only inhibits ZIKV infection, but can rescue ZIKV-induced growth and differentiation defects in human neuronal precursor cells and human fetal-like forebrain organoids [66].

### 2.4. Limitations of iPSC-Derived In Vitro Models

Even though iPSC-derived cardiomyocytes and neurons provide insights into the pathological mechanisms and high-throughput possibilities for drug screening, there still remain limitations for iPSC-derived disease models. It is challenging to mimic a complex intercellular communication of disease in adults due to the immature characteristic of iPSC-derived cells and the lack of intercellular communication among different cell types. Furthermore, many diseases are complex, often involving multiple genes and pathways. This is evident in diseases such as ASD which is a highly heterogenous disease. Individuals may carry a variety of gene mutations, but this may vary between individuals. Employing a single line of patient-derived cells may not provide a single hit compound for the spectrum of the disease, however, this will still aid in our understanding of this complex disorder.

The reproducibility of both the reprogramming and differentiation protocols has been frequently emphasized as an issue in most studies employing iPSCs. Overcoming these issues is essential for generating high-quality, widely applicable, high-yield iPSCs and derivative products [67,68,69,70]. Meanwhile, it brings out the likelihood that the iPSC-derived model might not work out before the standard of the protocols has achieved a certain level. A rigorous protocol ensures robust iPSC performance and consistency. On the other hand, those that are not as optimized may fail to generate fully representative models or platforms for studies requiring more stringent criteria to closely recapitulate in in vivo conditions.

In an impressive study, Kilpinen and colleagues analyzed copy number variation, gene expression, and proteomics in over 700 iPSC lines from 301 healthy donors. They showed that 5–46% of phenotypic variability in iPSCs is due to normal human genetic variation in both inter- and intraclone variability [71]. Matsa et al., 2016, suggest in their research that iPSC-derived cardiomyocytes retain the inter- and intravariation between and within cell lines. Following a rigorous reprogramming and standardized differentiation process, the iPSCs and the subsequent cardiac derivatives were demonstrated to maintain patient-specific genetic characteristics. This provides a potential platform for screening medicine specialized to patients, aiding the development of precision medicine [67]. Nevertheless, these data show the importance of appropriate controls, such as the generation of isogenic controls and rigorous protocols for the generation of iPSC-derivatives. Furthermore, the results obtained from a single line of iPSCs should be interpreted with caution.

### 2.5. Benefits and Limitations of High-Throughput Screening

High-throughput screening provides the opportunity to screen thousands of compounds in a single miniaturized screen, generally with the use of automation and is an essential tool in drug discovery. These screens are often performed in 864- and 1536-well plates and data acquisition is performed by an optical measurement. High throughput screens come with both benefits and challenges. Benefits include the use of lower volumes of reagents, a higher number of compounds screened on a single plate, resulting in reduced time and cost. However, there are challenges associated with miniaturization including the survival of cells plated in low numbers, evaporation of culture media due to low volume of reagents and accuracy in dispensing low volumes of compounds. The use of this technology in drug discovery has allowed the identification of many new compounds for the potential treatment of various diseases including Zika virus [66], SMA [51], as well as understanding cellular processes and developmental stages using genome editing strategies [72].

## 3. iPSC-Based Cell Therapy

The discovery of iPSC technology excited the scientific world for the potential of personalized cell therapy. However, many hurdles, such as tumorigenicity, immunogenicity, and heterogeneity, were quickly realized as potential problems that need to be overcome before these therapies can be fully integrated into a clinical setting. There are two approaches to iPSC-based cell therapy: autologous and allogeneic cell transplantation. Autologous transplantation utilizes a patient’s own iPSCs differentiated into target cells whereas, allogeneic transplantation uses iPSCs donated from a human leukocyte antigen (HLA) matched donor, similar to traditional organ donation. The HLA system regulates the immune system in humans. HLA matching between donor and recipient in organ transplantation and cell therapy is important to avoid immune-rejection by the host. Compared to ESCs, iPSCs are generated from patients and thus theoretically pose less risk of rejection after delivery into the same patients.

The first in-human clinical trial of iPSC-based cell therapy used an iPSC-derived retinal pigment epithelium (RPE) sheet that was implanted into a Japanese woman with age-related macular degeneration (AMD) in 2014 [73,74]. This patient did not experience any complications during surgery or after the surgery, however, the patient was required to wait over 10 months for the surgery and it cost nearly USD 1 million [75]. Since then, other clinical trials have been carried out resulting in clinically measurable visual improvements [76,77]. One currently ongoing phase 2/3 clinical trial at the Beijing University of Chinese Medicine is using autologous iPSC-derived cardiomyocytes to treat chronic heart failure with the aim of repairing injured myocardium. Their results from animal experiments confirmed the feasibility of intravenous myocardial cell transplantation. Another group in China, at Help Therapeutics, proposed using iPSC-derived cardiomyocytes to treat ischemic heart failures [78]. Although there are no efficacy data from their Phase 1/2 clinical trial at the moment, their preclinical study using rat models suggested that the implanted human cardiomyocytes improved heart function through cardiac remodeling [79]. Some of the main hurdles that are currently limiting this technology from moving forward are overcoming the inherent heterogeneity of the iPSC-derived cardiomyocytes, which are mostly composed of atrial, ventricular, and nodal-like cardiomyocytes; cell survival once injected into the damaged area, and obtaining the required cell number [80], all of which may lead to cardiac arrhythmias. There are currently many studies being undertaken to address all of the above issues in order to bring cardiac cell therapy to the clinic. However, autologous transplantation is an extremely time-consuming and expensive process, which hinders its ability to be readily applicable for acute progressive disorders.

More recently, iPSC-based cell therapies are switching from autologous transplantation to allogeneic transplantation [81]. Allogenic transplantations come with an increased risk for rejection as they are not the patient’s own cells. NK cells, which act as important immune cells, are widely used for transplantation [82,83]. NK cells can be differentiated from iPSCs and have proven to be equally effective as primary NK cells [84]. Moreover, antigen priming is not required for NK cells, therefore, it is not necessary to transplant with HLA matched NK cells [85]. FT500 is the first FDA-approved “off-the-shelf” product of homogeneous iPSC-NK from Fate Therapeutics in the USA, however, the safety and efficacy still need to be examined. Taking advantage of iPSCs, T cells can also be differentiated and served as a banked source for allogeneic transplantation [86,87], which decreases the cost and could be scaled-up to achieve “off-the-shelf” availability. Unlike chimeric antigen receptor (CAR) T cells from umbilical cord blood, CAR iPSC T cells are homogeneous due to derivation from one clonal engineered iPSC line. Moreover, these cells have been well established to avoid graft-versus-host disease by deletion of the TCR α chain [88]. However, the survival rate in patients is a concern [89] and it is necessary to test the safety and efficacy clinically.

Another study currently in Phase 1/2 clinical trials is using iPSC-derived dopaminergic progenitors as a cell-based therapy for Parkinson’s disease. In their preclinical mouse model, they showed behavioral improvement and no tumorgenicity or toxicity of the cells [90]. Other notable Phase 1 clinical trials in Japan includes an iPSC-derived cardiomyocyte sheet from a group comprising Cuorips Inc. and Osaka University [90]; their preclinical studies on ischemic cardiomyopathy pig models have shown improved cardiac function along with the identification of molecular factors to aid the electro-functional coupling between the transplanted cells and recipient heart [90]. The Phase 1 safety study currently has a total of 10 patients with ischemic cardiomyopathy and the trial is estimated to be complete by May 2023. Table 1 outlines the current (as of 2021) iPSC-based clinical trials.

### 3.1. HLA-Homozygous iPSC Banking

In order to avoid immunogenicity during allogenic cell therapy, HLA matching is required to avoid cell rejection. In allogeneic transplantation surgeries, HLA inconsistency between donors and recipients is the one of the main reasons for failure and rejection [91]. A previous study has identified over 9000 alleles within the HLA system [91] making it the most polymorphic genetic system in humans [92]. Although most of the pluripotent stem cells have low HLA expression [93], iPSC derivatives can trigger an immune response. This was confirmed in mice showing that iPSC-derived cell implantation leads to cellular apoptosis and T cell proliferation [94]. Therefore, the iPSC lines used for cell therapy and the recipient should at least have a certain degree of HLA matching. iPSCs reprogrammed from donors with similar homozygous HLA haplotypes have the potential to be used for allogeneic cell therapy, however, finding a match can be time consuming, difficult, and financially challenging. Therefore, HLA banking can potentially benefit a large proportion of the population by having a selection of HLA homozygous cells readily available. It has been shown that a small number of homozygous HLA haplotypes circumvents prospective typing of a large number of individuals [95]. The iPSCs derived from these donors are HLA compatible with a large proportion of the population. An ample number of iPSCs can be prepared and banked, which can help reduce the cost and time of the iPSC manufacturing process. In 2016, Sugita et. al. showed that transplantation of RPE cells caused minimal immune reactivity in MHC, the monkey HLA equivalent, matched monkeys [96]. This was further confirmed in a clinical trial study with five recruited HLA-matched patients who received HLA-matched RPE cells [97]. In all five cases, survival of transplanted cells was observed one year later, without the need for systemic immune suppression. In another proof of concept study, transplantation of MHC-matched iPSC-derived dopaminergic neurons in non-human primates has been reported. Morizane et al. and Kikuchi et al. both showed that an improvement of engraftment and reduced immune reaction were observed using MHC matching of iPSC-derived dopaminergic neurons in unlesioned non-human primates [98,99]. In the field of cardiomyocyte transplantation, Kawamura et al. found that, although MHC-matched cardiomyocytes reduced immunogenicity, appropriate immunosuppression was also required to ensure successful engraftment [100]. Shiba et al. demonstrated that MHC-matched cardiomyocytes under immunosuppression treatment had successfully restored cardiac function and survived after 12 weeks transplantation [101].

Even though disease modeling is not the primary aim for HLA iPSC banking, these iPSCs can also be used to clarify the disease mechanisms of HLA-associated autoimmune and infectious disease [102]. Previous studies have shown the correlation between HLA complex and population-specific adverse drug reactions. In 2002, Mallal et al. showed the most prevalent Southeast Asian population HLA complex, HLA-B*57:01, correlated with abacavir sensitivity [103]. Furthermore, several studies have found that HLA-B*58:01, which was expressed relatively more frequently in the Chinese population (10–15%) compared to the Caucasian population (0.8%) is strongly associated with allopurinol sensitivity and cause Stevens–Johnson Syndrome [104]. Adverse drug reactions may vary depending on the genetic background of the recipients; however, these adverse drug reactions may not be detectable during conventional drug development or the clinical trial process. In a recent study, Huang et al., 2022, have reported the generation of an HLA-homozygous iPSC bank by screening 1000 healthy donors in Taiwan, and established the first drug screening model representing the Han Chinese population in Taiwan [105]. The iPSC derivatives were also used to assess the cardiotoxicity and neurotoxicity.

The benefit of establishing an HLA homozygous iPSC bank is that the HLA haplotypes of the iPSCs derived from a small number of donors can match a large number of the population. For instance, an HLA iPSC bank with iPSCs derived from 55 donors can be used to perform iPSC-based cell therapy for 80% of the Japanese population, a country with limited HLA variability [106]. To reach the same level of patient coverage in nations with mixed races, an HLA iPSC bank with a larger number of donors is required, due to the higher HLA variability within the population. Since HLA haplotype overlapping can occur in different countries, it is important that donors are screened for homozygous HLA haplotypes and selected with international collaboration. Several studies have been undertaken to calculate the number of iPSC donors required to cover enough of the population in different countries. For example, Taylor et al. showed that 93% of the UK population can be covered with an HLA iPSC bank of 150 iPSCs derived from selected donors with HLA homozygous haplotypes [95]. Meanwhile, it has been suggested that 140 iPSC lines can cover 90% of the Japanese population [107]. In regions such as California, USA, a population with highly diverse HLA haplotypes, approximately 80 iPSC lines from donors with different ancestry groups will be required to reach 50% coverage of the population [108].

Recently, researchers throughout the world have begun to invest in establishing HLA homozygous iPSC banks. The Center for iPS Cell Research and Application (CiRA) has reported the establishment of three clinical grade HLA iPSCs, homozygous at the allele level of HLA-A, HLA-B and HLA-DR, which can cover 32% of the Japanese population [109]. The CHA Stem Cell Institute in Korea has announced the repurposing of a cord blood bank to select donors and generate 10 HLA homozygous iPSCs, conditional to three HLA loci (HLA-A, HLA-B, and HLA-DR). These iPSCs can cover 41% of the Korean population [110]. Furthermore, 13 HLA homozygous iPSCs, generated by the Taiwan Human Disease iPSC Service Consortium, can cover about 16% of the Taiwanese population based on the HLA-A, HLA-B, and HLA-DRB1 alleles, and was calculated to further represent at least 4.77 million of the world population [105].

Development of an HLA homozygous iPSC bank in countries with high HLA variability may require more time and money to achieve wider coverage. For instance, 631 HLA homozygous iPSCs would be required to cover 90% of the Spanish population, while less than 200 iPSCs are required to reach the same coverage in Korea [111]. Reducing the required number of matching HLA alleles may help to reduce the number of iPSC lines required to cover most of the population in countries with high HLA variability. However, the degree of HLA matching between donors and the population remains controversial, and more research needs to be done to determine the level of HLA matching stringency during screening of the HLA homozygous donors.

### 3.2. Genome Edited iPSCs

In order to overcome some of the rejection issues associated with HLA specificity, genome editing tools, especially the recently developed short palindromic repeats (CRISPR)/ CRISPR-associated protein 9 (Cas9) system, have opened up many new possibilities in the application of iPSCs including fundamental gene expression studies, disease modeling and stem cell therapy. As mentioned previously, the generation of HLA-homozygous iPSC banks that could match the majority of the intended recipients in a population could be difficult and time consuming as HLA genotypes are highly variable within a population, especially in multiethnic countries [95]. For example, to acquire enough cell lines to represent around 90% of the entire Japanese population, at least 150,000 donors must be screened [107]. As an alternative, to reduce the screening costs, several scientists have suggested using gene editing to create HLA homozygous cells. With the CRISPR/Cas9 system, generation of these pseudo-homozygous iPSCs is possible through specific and accurate allele editing of available heterozygous iPSCs [112].

On the other hand, generation of a “universal” donor stem cell could prove to be more time and cost efficient as long as they are hypoimmunogenic, preventing graft-versus-host disease and even death in recipient patients [86]. A group has reported that the ablation of the B2M gene which encodes β2-microglobulin, a protein required for HLA class I presentation on the cell surface, could negate the cytotoxic effects of CD8^+^ T cells, which could ease iPSC-derived tissue transplantation [113,114]. Other groups have also reported that knockout of HLA-B could result in less immunogenicity [115]. However, the removal of certain HLA genes could trigger the activation of the NK cells [116,117]. The CRISPR/Cas9 system has since been employed to accurately disrupt both the HLA-A and HLA-B alleles while retaining HLA-C genes in iPSCs, which could inhibit CD8^+^ T cells while not triggering NK cell activity in allogenic transplantations [112]. Scientists have also discovered that iPSCs can have improved immune escape by transducing the NK-inhibitory ligand-scHLA-E genes in addition to the knocking out of the reported HLA-I- and HLA-II-related genes and NK cell-activating ligand gene (B2M gene, CIITA gene and PVR gene, respectively) [118]. Although the CRISPR/Cas9 genome-editing tool can inhibit immune rejection in allogenic stem cell therapies, it is also a double-edged sword as HLA-edited cells, especially iPSCs, would have an increased risk of tumorigenicity due to their acquired ability to escape immune surveillance of the host. However, fully differentiated, hypoimmunogenic iPSC-derived T cell treatment models are able to reap the benefits of CRISPR/Cas9 system without the side effects of tumorgenicity [87]. For example, one of the most promising hematological anticancer therapies is the chimeric antigen receptor (CAR) T cell therapy. Previously reported limitations of the CAR T cell therapy showed that immunocompromised patients may not have enough T cells in their bodies and patients with late stage acute diseases would not be able to endure the lengthy manufacturing process of autologous CAR T. These issues can be overcome with readily available “universal” clinical-grade CAR T cells. In other words, iPSCs from healthy donors could be genetically modified with CRISPR/Cas9 system to be free of allogenic factors and could be mass-produced in advance to be clinically applicable to all patients without delay. Modification of endogenous TCR to eliminate the alloreactivity of transplanted anti-tumor CAR T has already been demonstrated.

Besides the modification of the TCR, CRISPR/Cas9 genome engineering’s usefulness can also be seen in human disease modeling. Precision editing of alleles in a human cell has drastically improved the understanding of genotypic and phenotypic effects in gene therapy due to the lower off-target rates [119,120,121]. iPSCs in particular, benefit the most from CRISPR/Cas9-assisted disease modeling due to their unlimited cell source and ability to retain edited genotypes that recapitulate the phenotypes of cells and tissue of the diseased patients [122]. The generation of isogenic controls, whereby correcting gene mutations back to wild-type, provide the best controls for disease models. Such disease models could also be used in combination with in silico approaches for high-throughput drug discovery and drug development studies [72].

Despite CRISPR/Cas9 genome engineering’s robust and diverse utility, there are still issues to overcome. For example, off-target cleavages of the Cas9 nucleases that cause oncogenic mutations in the genome have been reported extensively [123,124]. Off-target genome editing is often hard to detect as it may vary among different cell types and species, along with different frequency influenced by the abundance of Cas9 proteins in the reaction, structure of sgRNA, and cellular state [125]. Current strategies to prevent off-target activities include screening of highly predicted off-target sections, fusion of Cas9 proteins with Fokl nucleases, or generation of improved guide RNAs and Cas9 variants [126,127,128].

## 4. Future Perspectives

The possibility to model diseases and their use in cell therapy have made iPSCs an exciting tool for scientists. Although protocols for the generation of 3D models have become more available, the output and data analysis are highly complex and time consuming. Their use in high-throughput drug screening and drug discovery is still limited. Therefore, there is a need to develop reliable protocols for high-throughput generation of various organoids as well as assays for the detection of toxicity and functionality. Furthermore, organoids only represent fetal-like structures, thus the culturing conditions, time, and protocols may need further development to generate more mature, adult-like equivalents. Despite this, iPSCs represent an exciting and valuable tool for the generation of disease models, drug discovery, and play an important role in cell-based therapy.

## Figures and Tables

**Figure 1 cells-11-01853-f001:**
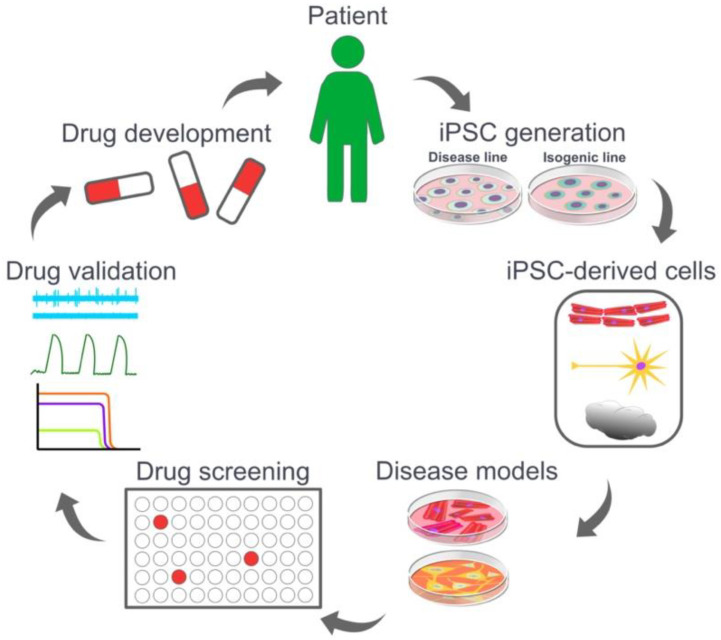
Disease modeling and drug screening using patient-specific induced pluripotent stem cells.

**Table 1 cells-11-01853-t001:** Current iPSC-based clinical trials (as of 2021).

Location	Company	Disease	Cell Type	Clinical Phase	Clinical Trial Identifier
Australia, United Kingdom	Cynata Therapeutics Limited	Graft vs. host disease	iPSC-derived mesenchymal stem cell	Phase 1	ClinicalTrials.gov: NCT02923375
United States	Fate Therapeutics	Cancer	iPSC-derived Natural Killer (NK) cell	Phase 1	ClinicalTrials.gov: NCT03841110
China	Beijing University of Chinese Medicine	Chronic heart failure	iPSC-derived cardiomyocytes	Phase 2/3	ClinicalTrials.gov: NCT03759405
	Help Therapeutics	Heart failure	iPSC-derived cardiomyocytes	Phase 1/2	ClinicalTrials.gov: NCT03763136
Japan	Kyoto University Hospital	Parkinson disease	iPSC-derived dopaminergic progenitors	Phase 1/2	ICTRP: JPRNUMIN000033564
	Osaka University, Cuorips Inc.	Myocardial ischemia	iPSC-derived cardiomyocytes sheet	Phase 1	ClinicalTrials.gov: NCT04696328

## Data Availability

Not applicable.

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
