# Peer review of "Utility of iPSC-Derived Cells for Disease Modeling, Drug Development, and Cell Therapy"

_cells, 2022, doi:10.3390/cells11111853_

Round 1
Reviewer 1 Report
Nicholson et al provide an informative review of disease modelling, drug development and stem cell therapies. Overall the content is informative, accurate and compelling. Several minor issues may enhance the impact and quality of the overall text;
1) the initial sections on disease modelling and drug screening are not as well researched and discussed as the stem cell therapies sections. Greater depth and discussion may enhance the points, as well as reorganization of certain sections.
For example, most studies described are basic research with a single drug treatment – as a focus of the paper is drug discovery/development, a section describing the benefits and recent high throughput screens may be desirable. You describe a screen of 50,000 compounds for fragile X, but there are numerous other screens published, enough for a valuable insight into the benefits, challenges and results of iPSC-derived HTS. This is also the case for toxicity screening where high throughput methods with iPSC derived cell types are now the standard for novel anti-cancer drugs (and becoming widespread for neurological disorder treatments), and more examples of this are available than reference 69 (line 210).
Cardiomyocytes and neuron sections;
Both sections could be strongly enhanced by describing the in vivo use of any drugs isolated from iPSC studies – if there are any drugs isolated or validated in iPSC that have translated well to animal models/ the clinic. This would support the claims that iPSC-derived studies have benefits over traditional animal models and some (or a single) success story would be of great interest to the scientific community.
Limitations of iPSC models section – ASD is highly heterogeneous, how do you perform effective drug discovery when 1000+ genes contribute to the disorder (do you find network hubs, or attempt to recover a specific facet of neurological/cardiac function independent of risk genes – perhaps the papers cited in these sections would provide a rational approach? . This is particularly worthy of discussion, as isogenic lines (often employed for drug screening) may suffer similarly to animal models in failing to recapitulate genetic complexity, whilst large patient cohorts stifle drug discovery. Extra discussion may be of interest to the community.
Organoids section; Inclusion of several papers may be relevant and of interest to the scientific community; PMID: 32989314, PMID: 22120178, PMID: 34548630 and many more - all describe organoids of neurodevelopmental disorders and perform drug treatments. Other CNV and disease model citations would be relevent to include to enhance the organoid section.
2) Several terms are not discussed before first mention, and limit the understanding outside of isolated fields. For example; apheresis (line 29). No definition of autologous vs allogeneic transplantation. HLA is not defined in the text and HLA matching/banking is not described in detail. Some more explanation may aid readers.
3) several inaccuracies, likely textual mistakes, could be amended; Line 42 - 45 “although many of these offer specific advantages…. They are not of human origin” – many immortal cell lines (e.g. HEK293, HELA) and primary tissues (e.g. HUVEC) are from human tissue and so this sentence perhaps is not completely accurate. “ESC and animal models… ethical issues… hinders their utility” – Every drug ever brought to the market has employed extensive animal testing, despite ethical issues. This is hardly a barrier to utility in animal models. Perhaps ESC suffer from this limitation, and so animal and ESC should be separated. Please check the accuracy of such broad statements throughout and amend or provide relevent citation.
4) An effort to make the relevent conclusions would aid the readers appreciation of the implications of the cited studies. E.g. What was the outcome of RPE implantation (line 276)- was it successful? The citations in table 1 are not discussed in the text and it would benefit the reader to know the clinical trial reference (NCT# (simply citing clinicaltrials.gov does not aid the reader)) numbers and any results (if available) for these technologies (e.g. success in phase 1 indicated safety, Phase 2/3 study shows preliminary success in xxx endpoint). If no efficacy data is available for clinical trials, perhaps expand the discussion of preclinical success of the indicated technologies, particularly with regard to existing/competing technologies.
5) Additional subheadings throughout would aid the reader in digesting the material.
Author Response
Nicholson et al provide an informative review of disease modelling, drug development and stem cell therapies. Overall the content is informative, accurate and compelling. Several minor issues may enhance the impact and quality of the overall text;
1) The initial sections on disease modelling and drug screening are not as well researched and discussed as the stem cell therapies sections. Greater depth and discussion may enhance the points, as well as reorganization of certain sections. For example, most studies described are basic research with a single drug treatment – as a focus of the paper is drug discovery/development, a section describing the benefits and recent high throughput screens may be desirable. You describe a screen of 50,000 compounds for fragile X, but there are numerous other screens published, enough for a valuable insight into the benefits, challenges and results of iPSC-derived HTS. This is also the case for toxicity screening where high throughput methods with iPSC derived cell types are now the standard for novel anti-cancer drugs (and becoming widespread for neurological disorder treatments), and more examples of this are available than reference 69 (line 210).
We thank the reviewer for their comments. We have included various screens that utilize high throughput screening throughout the paper including a section dedicated to the benefits and limitations of high-throughput screening.
Page 7, lines 288-301
2.5. Benefits and limitations of high-throughput screening
High-throughput screening provides the opportunity to screen thousands of compounds in a single miniaturized screen, generally with the use of automation, and is an essential tool in drug discovery. These screens are often performed in 864- and 1536-well plates and data acquisition is performed by optical measurement. High throughput screens come with both benefits and challenges. Benefits include the use of lower volumes of reagents, and a higher number of compounds screened on a single plate, resulting in reduced time and cost. However, there are challenges associated with miniaturization including survival of cells in plated in low numbers, evaporation of culture media due to the low volume of regents, and accuracy in dispensing low volumes of compounds. The use of this technology in drug discovery has allowed the identification of many new compounds for the potential treatment of various diseases including Zika virus [66], SMA [51], as well as understanding cellular processes and developmental stages using genome editing strategies [72].
Cardiomyocytes and neuron sections;
Both sections could be strongly enhanced by describing the in vivo use of any drugs isolated from iPSC studies – if there are any drugs isolated or validated in iPSC that have translated well to animal models/ the clinic. This would support the claims that iPSC-derived studies have benefits over traditional animal models and some (or a single) success story would be of great interest to the scientific community.
We thank the reviewer for their suggestion. We have included a study that first identified a compound in a screen using iPSC-derived cells, which was then taken forward for clinical trials. We agree this has enhanced the manuscript.
In Neuron subheading; page 5, line 198-202.
Bringing drugs from the lab to clinical trials is the end goal of many drug screening studies. Ohuchi et al (2006) first used SMA patient iPSC-derived motor neurons and astrocytes to show the role of thyrotropin-releasing hormone analog for the treatment of SMA, which has been taken to clinical trials and showed the potential efficacy of this drug for the treatment of SMA [44].
Limitations of iPSC models section – ASD is highly heterogeneous, how do you perform effective drug discovery when 1000+ genes contribute to the disorder (do you find network hubs, or attempt to recover a specific facet of neurological/cardiac function independent of risk genes – perhaps the papers cited in these sections would provide a rational approach? . This is particularly worthy of discussion, as isogenic lines (often employed for drug screening) may suffer similarly to animal models in failing to recapitulate genetic complexity, whilst large patient cohorts stifle drug discovery. Extra discussion may be of interest to the community.
We thank the reviewer for their thoughtful suggestion. We have added further discussion on more complex diseases such as ASD and the difficulty in identifying hit compounds for this type of disease.
In Limitations of iPSC-derived in vitro models section; page 6, line 260-265.
“Furthermore, many diseases are complex, often involving multiple genes and pathways. This is evident in diseases such as ASD, a highly heterogeneous disease. Individuals may carry a variety of gene mutations, but this may vary between individuals. Employing a single line of patient-derived cells may not provide a single hit compound for the spectrum of the disease, however, this will still aid in our understanding of this complex disorder.”
Organoids section; Inclusion of several papers may be relevant and of interest to the scientific community; PMID: 32989314, PMID: 22120178, PMID: 34548630 and many more - all describe organoids of neurodevelopmental disorders and perform drug treatments. Other CNV and disease model citations would be relevent to include to enhance the organoid section.
We appreciate this comment from the reviewer and have added more examples within the organoid sections of the manuscript.
In Cardiac organoid section; page 5, line 222-227.
“Another group, which previously developed a high-throughput bioengineered human cardiac organoid platform [59], has generated organoids for the purpose of identifying pro-proliferative compounds which were followed by validation and functional screening to identify hit compounds with side effects on contractility [56]. They used these organoids to further validate the functional pathways involved in the proliferative effects.”
In Cardiac organoid section; page 5, line 227-229.
“Self-assembling human cardiac organoids have also been used to show the role of BMP4 and Activin A to improve heart organoid chamber formation and vascularization [60].”
In Cardiac organoid section; page 6, line 233-234.
Cardiac organoids have also been used to screen a panel of environmental toxins, showing the utility of these models for the use of toxicity screening [62].
In Neuronal organoid section; page 6, line 232-236.
“Another study using 100-day cerebral cortical organoids showed that organoids derived from patients with 22q11.2 deletion syndrome had deficits in spontaneous neuronal activity and calcium signaling. The authors showed that antipsychotics could restore the functional defects found in 22q11.2 deletion syndrome providing further justification for the use of iPSC-derived organoids as a relevant model for disease modeling [65].”
2) Several terms are not discussed before first mention, and limit the understanding outside of isolated fields. For example; apheresis (line 29). No definition of autologous vs allogeneic transplantation. HLA is not defined in the text and HLA matching/banking is not described in detail. Some more explanation may aid readers.
We thank the reviewer for bringing this to our attention. We have defined more technical terms such as apheresis and HLA in the manuscript.
In Introduction; page 1, lines 27-31.
“The capacity of iPSCs for scalable production of diverse cell types benefits researchers by reducing obstacles, such as procedures like apheresis, which involves separating specific components from the blood and returning the remaining components to the donor’s circulation, whilst obtaining human cells.”
In iPSC-based cell therapy; page 7, line 306-312.
“There are two approaches to iPSC-based cell therapy: autologous and allogeneic cell transplantation. Autologous transplantation utilizes a patient’s own iPSCs differentiated into target cells whereas, allogeneic transplantation uses iPSCs donated from a human leukocyte antigen (HLA) matched donor, similar to traditional organ donation. The HLA system regulates the immune system in humans. HLA matching between donor and recipient in organ transplantation and cell therapy is important to avoid immune-rejection by the host.”
3) several inaccuracies, likely textual mistakes, could be amended; Line 42 - 45 “although many of these offer specific advantages…. They are not of human origin” – many immortal cell lines (e.g. HEK293, HELA) and primary tissues (e.g. HUVEC) are from human tissue and so this sentence perhaps is not completely accurate. “ESC and animal models… ethical issues… hinders their utility” – Every drug ever brought to the market has employed extensive animal testing, despite ethical issues. This is hardly a barrier to utility in animal models. Perhaps ESC suffer from this limitation, and so animal and ESC should be separated. Please check the accuracy of such broad statements throughout and amend or provide relevent citation.
We thank the reviewer for bringing this to our attention. We have amended this in the manuscript to ensure accuracy and clarity in our statements.
In Drug discovery using iPSC-derived disease models section; page 2, line 44-45.
“Although many of these models offer specific advantages, the underlying issue is that they often do not accurately model human diseases.”
In Drug discovery using iPSC-derived disease models section; page 2, line 45-46.
“Furthermore, the use of ESCs raises an inextricable ethical issue, which largely hinders their utility.”
4) An effort to make the relevent conclusions would aid the readers appreciation of the implications of the cited studies. E.g. What was the outcome of RPE implantation (line 276)- was it successful?
We thank the reviewer for their thoughtful critique. We have included conclusions and results from the RPE implantation clinical trials.
In iPSC-based cell therapy section; page 7, line 319-320.
Since then, other clinical trials have been carried out resulting in clinically measurable visual improvements [70,71].
The citations in table 1 are not discussed in the text and it would benefit the reader to know the clinical trial reference (NCT# (simply citing clinicaltrials.gov does not aid the reader)) numbers and any results (if available) for these technologies (e.g. success in phase 1 indicated safety, Phase 2/3 study shows preliminary success in xxx endpoint). If no efficacy data is available for clinical trials, perhaps expand the discussion of preclinical success of the indicated technologies, particularly with regard to existing/competing technologies.
We thank the reviewer for their comment. We have added more details information on the clinical trials where available. We have also included the NCT number in Table 1.
In the iPSC-based cell therapy section; page 7, line 323-328
Their results from animal experiments confirmed the feasibility of intravenous myocardial cell transplantation. Another group in China, at Help Therapeutics, proposed using iPSC-derived cardiomyocytes to treat ischemic heart failures [73]. Although there are no efficacy data from their Phase ½ clinical trial at the moment, their preclinical study using rat models suggested that the implanted human cardiomyocytes improved heart function through cardiac remodeling [74].
In the iPSC-based cell therapy section; page 8, line 356-362
Other notable Phase 1 clinical trials in Japan include iPSC-derived cardiomyocytes sheet from a group comprised of Cuorips Inc. and Osaka University [85], their preclinical studies on ischemic cardiomyopathy pig models has shown improved cardiac function along with the identification of molecular factors to aid the electro-functional coupling between the transplanted cells and recipient heart [85]. The Phase 1 safety study currently has a total of 10 patients with ischemic cardiomyopathy and the trial is estimated to be complete by May 2023.
5) Additional subheadings throughout would aid the reader in digesting the material.
We have included new subheadings in the neuron section to help better organize the material for the reader.
In Neuron subheading; pages 3 and 4.
2.2.1. Neurodegenerative diseases
2.2.2. Neurodevelopmental diseases
Reviewer 2 Report
This is a well written review which contributes to a large body of information on the utility of iPSc and provides a nice summary.
The overall comment is that the title "Utility of iPSC-derived cardiomyocytes and neurons for disease 2 modeling, drug development, and cell therapy" only goes so far in describing the content.
The authors spend a lot of time on describing immune cells especially in cell transplantation section. It does not seem to be very relevant to the title.
The rest of the comments are mostly very minor:
1. Page 3 line 83: "The left ventricle of the heart enlarges diastolic dysfunction, and ventricular arrhythmogenesis".
Perhaps the authors meant: "The heart enlarges causing diastolic dysfunction and..."
2. Page 3 line 124: AD is not "another type", it is "the commonest type of..."
3. Page 3 line 125: What do they specifically mean by "many phenotypes"?
4. Page 4 line 137. Do the authors mean AD rather than AZ?
5. Do they mean "most prevalent neurodegenerative disorder after AD"?
6. " PD symptoms" not syndromes!
7. Page 4 line 144: not "which consist" but instead "consistent with"
8. Page 4 line 145 "parkin". Parkin is a gene and needs to be at least capitalised. The authors might want to consider italicising all genes according to the gene nomenclature.
9. Page 4 line 150: do they mean "researchers"?
10. Page 4 line 157: do they mean "complex" rather than "multiplex"?
10 Page 4 line 158: do they mean "pathology" rather than "pathogen"?
11. Page 4 line 175. Do they mean SMN1?
12. Page 6 line 235: "Limitations" not "limitation"
13. Page 9 line 374. Who is the "Japanese scientist"?
Author Response
Reviewer 2
This is a well written review which contributes to a large body of information on the utility of iPSc and provides a nice summary.
The overall comment is that the title "Utility of iPSC-derived cardiomyocytes and neurons for disease 2 modeling, drug development, and cell therapy" only goes so far in describing the content.
The authors spend a lot of time on describing immune cells especially in cell transplantation section. It does not seem to be very relevant to the title.
We thank the reviewer for the comment. We agree with the reviewer and we have amended the title of the manuscript to “Utility of iPSC-derived cells for disease modeling, drug development, and cell therapy ”.
Revised manuscript: page 1, line 2
“Utility of iPSC-derived cells for disease modeling, drug development, and cell therapy ”.
The rest of the comments are mostly very minor:
We thank the reviewer for their comments below. We have amended each of the comments as suggested.
- Page 3 line 83: "The left ventricle of the heart enlarges diastolic dysfunction, and ventricular arrhythmogenesis".
Perhaps the authors meant: "The heart enlarges causing diastolic dysfunction and..."
In Cardiomyocytes subheading; page 3, line 85-87.
“The heart enlarges causing diastolic dysfunction and ventricular arrhythmogenesis which are the main pathological features of hypertrophic cardiomyopathy (HCM).”
- Page 3 line 124: AD is not "another type", it is "the commonest type of..."
In Neurons subheading; page 3, line 127-128.
“Alzheimer's disease (AD) is the most common type of neurodegenerative disease that affects memory and behavior.”
- Page 3 line 125: What do they specifically mean by "many phenotypes"?
In Neurons subheading; page 4-5, line 128-130.
“IPSC-derived neurons display many of the cellular phenotypes found in AD patients often associated with amyloid precursor protein (APP) [44], presenilin [45], or SORL1 mutation [46].”
- Page 4 line 137. Do the authors mean AD rather than AZ?
In Neurons subheading; page 4, line 139-140.
“They then employed this model to screen for potential therapeutics for the treatment of AD.”
- Do they mean "most prevalent neurodegenerative disorder after AD"?
In Neuron subheading; page 4, line 143.
“Parkinson's disease (PD) is the second most prevalent neurodegenerative disease after AD with a progressive loss of ventral midbrain dopaminergic neurons (vmDAns).”
- " PD symptoms" not syndromes!
In Neurons subheading; page 4, line 145.
“PD symptoms include bradykinesia, impaired balance, and rigid muscle.”
- Page 4 line 144: not "which consist" but instead "consistent with"
In Neuron subheading; page 4, line 147.
“SNCA triplication patient iPSCs-derived dopaminergic neurons showed α-synuclein protein over accumulation consistent with pathological phenotype in PD patient.”
- Page 4 line 145 "parkin". Parkin is a gene and needs to be at least capitalised. The authors might want to consider italicising all genes according to the gene nomenclature.
We thank the reviewer for their comment. We have italicized all genes throughout the manuscript.
In Neuron subheading; page 4, line 148.
“PD patient iPSCs-derived dopaminergic neurons harboring Parkin mutation displayed decreased microtubule stability, increased oxidative stress, reduced dopamine uptake, and increased spontaneous dopamine release.”
- Page 4 line 150: do they mean "researchers"?
In Neuron subheading; page 4, line 152-154.
Identification of therapeutic candidates for PD is the focus of many research labs and private companies worldwide and has yielded some potential targets.
- Page 4 line 157: do they mean "complex" rather than "multiplex"?
In Neuron subheading; page 4, line 173.
“ASD is a complex developmental disease of the brain with complex etiology or genetic pathology, characterized by restricted/repetitive behaviors and social communication disturbance.”
10 Page 4 line 158: do they mean "pathology" rather than "pathogen"?
In Neuron subheading; page 4, line 174.
“ASD is a complex developmental disease of the brain with complex etiology or genetic pathology, characterized by restricted/repetitive behaviors and social communication disturbance.”
- Page 4 line 175. Do they mean SMN1?
In Neurons subheading; page 5, line 191
“The most common form of SMA is caused by defects in survival motor neuron (SMN1) expression.”
- Page 6 line 235: "Limitations" not "limitation"
Page 6, line 254
“Limitations of iPSC-derived in vitro models”
- Page 9 line 374. Who is the "Japanese scientist"?
In the HLA-homozygous iPSC banking subheading; page 10, line 427-428
“Meanwhile, it has been suggested that 140 iPSC lines can cover 90% of the Japanese population”